# Reinforcement learning for hierarchical proof generation in Lean 4

## Abstract

Scaling formal theorem proving to more complex problems requires more efficient infer-ence methods than standard whole-proof generation, which struggles with longer proofs due to exponentially decreasing success rates with increasing proof length. In this work, we systematically study how online reinforcement learning can be coupled with a hier-archical (lemma-based) style of proof generation, that has recently gained popularity for inference-time methods. We show a fruitful interaction in two ways: reinforcement learn-ing allows to train proof decomposition policies successfully, and hierarchical inference allows to overcome plateaus in reinforcement learning as its richer distribution favors exploration and generated lemmas can be understood as an online synthetic data generation technique. Overall, hierarchical inference trained with reinforcement learning produces strong numbers on evaluations, even at the 7B parameter scale, and outperforms standard whole-proof generation setups in terms of sample efficiency and scalability, making it a suitable technique for necessarily data-constrained formalization research efforts.

## 1 Introduction

With the availability of increasingly powerful pretrained large language models, machine learning for formal theorem proving has made rapid progress in the past years. At the same time, modern models' higher accuracy and stability over long generations made new paradigms applicable for proof generation. While in early works, *tree search* with line-level tactic generation models that interact with the environment state provided by interactive theorem provers dominated (e.g. (Polu & Sutskever, 2020; Han et al., 2021; Lample et al., 2022; Wu et al., 2024a; Gloeckle et al., 2024)), more recent works moved to *whole-proof generation* where the model simply outputs the entire proof in one go (Jiang et al., 2023; Xin et al., 2024; Ren et al., 2025; Dong & Ma, 2025; Wang et al., 2025) and internalizes the *world modeling* of predicting state transitions in the environment.

Due to the autoregressive nature of text generation from language models, long-form generation results in a similarly diverse proof attempt distribution as tree search, making the combination of large-scale sampling and exact verification that proof assistants provide a viable alternative to tree search that, moreover, is more aligned with computational and hardware constraints.

However, full-proof generation suffers from *poor scalability* in terms of proof complexity: assuming a success rate of $p \in [0, 1]$ for an elementary step in a proof, an $n$-step proof would have an overall success rate of $p^n$, i.e. an exponential shrinking with respect to proof length. While this is a very rough modeling as $p$ might depend on $n$, it is quite illustrative of this lack of scalability.

To address this shortcoming, *decomposition and hierarchical recursion* is a natural method to apply to such complex construction tasks: problems are broken down into individual parts which are solved individually before re-assembling the overall solution from them. In the context of machine learning for formal theorem

proving, decomposition corresponds to *lemma-based proving*, or *sketch then prove*, where a proof *sketch* consisting of several lemmas is generated before the lemmas are proved individually.

Language modeling for formal theorem proving faces the additional challenge that training data is scarce. For instance, in the Lean 4 theorem proving language, "only" of the order of hundreds of millions of tokens exist, while large language models are routinely trained on trillions and large reinforcement learning runs produce billions of tokens Although this has been partially mitigated for Olympic-style mathematics through the widespread use of synthetic data, the situation is even worse for cutting-edge mathematical research area where available data is, by definition, very scarce. Hence, reinforcement learning has become the dominant method for training leading language models for theorem proving.

While past works frequently either focused on hierarchical approaches to neural theorem proving in an inference-only or synthetic data generation setup, or on reinforcement learning but with fairly standard whole-proof generation methods (see Section 6), in this work we study how reinforcement learning can be applied to hierarchical proving, choosing the Lean 4 proof assistant as a well-studied and convenient testbed.

Our findings are as follows:

- Hierarchical proof generation can easily be trained with reinforcement learning with minimal amounts of supervised finetuning data specialized for the respective regime. Unlabeled reinforcement learning dataset size likewise can be several orders of magnitudes below that of current state-of-the-art methods.
- Hierarchical proof generation performs favorably compared to standard whole-proof generation, showing higher sample efficiency and plateauing later in reinforcement learning runs. In particular, they exhibit a cumulative solve rate that keeps increasing long after classical whole proof generation methods based on Group Relative Policy Optimization (GRPO) training plateau.
- The resulting models show convincing numbers both in the small and high sample regimes.

## 2 BACKGROUND

### 2.1 SATURATION IN REINFORCEMENT LEARNING

Unlike in language model pretraining where performance is predictably governed by *scaling laws* (Kaplan et al., 2020; Hoffmann et al., 2022; OpenAI, 2023), reinforcement learning with language models rather appears to follow a "saturation model": with increasing RL compute budget, evaluation performance shows diminishing returns and even degrades later in training (Cui et al., 2025; Yue et al., 2025). Policy entropy (or KL divergence to the initial policy) can be seen as an *optimization budget* that the RL algorithm spends until no further improvements can be made (Gehring et al., 2024; Walder & Karkhanis, 2025).

We establish the following criteria for identifying a **plateau** in a reinforcement learning run: **(i)** policy entropy almost vanishes, **(ii)** almost no new problems from the training set are being solved and **(iii)** evaluation performance stagnates or declines.

In Section 5.2, we establish that standard reinforcement learning plateaus early for a best-in-class model for Lean.

### 2.2 TOKEN CRISIS IN LLM TRAINING

With compute budgets projected to increase exponentially over the upcoming years, language model pretraining is predicted to hit the "token wall" when available organic data runs out compared to optimal scaling requirements (Xue et al., 2023; Gadre et al., 2023). Beyond that point, synthetic data, post-training and reinforcement learning are expected to take an even more prominent role (Villalobos et al., 2024).

Language modeling for Lean can be seen as a case study for this situation: with organic tokens only ranging at the 100M scale (Aram H. Markosyan, 2024; Wu et al., 2024b), state of the art models already heavily rely on synthetic data and reinforcement learning (Lin et al., 2025; Ren et al., 2025; Dong & Ma, 2025) for additional data in the tens of billions of tokens scale, dwarfing the organic data by three orders of magnitude.

While our analysis will be focused on language models for Lean specifically, this study can also be understood as an early data point on best practices for *data-constrained* language modeling.

### 2.3 TECHNIQUES TO MITIGATE SATURATION

To delay saturation effects in reinforcement learning, various techniques have been proposed, reviewed here with a focus on theorem proving in Lean:

- **Expert iteration**, where data is collected from the current policy $\pi_t$, merged with data from previous iterations $\pi_s$, $s < t$, and used to finetune the next iteration $\pi_{t+1}$ from the initial policy $\pi_0$ (Anthony et al., 2017) (see also (Polu & Sutskever, 2020; Lin et al., 2025; Ren et al., 2025; Wu et al., 2024a; Ying et al., 2024)) Expert iteration allows using large finetuning learning rates of the order of $10^{-5}$, as opposed to the smaller RL learning rates of the order of $10^{-7}$.

- **Scaling training datasets** in order to reduce epoching and overfitting effects.

- **Pass@k training** and other alternative reward formulations that favor diversity by design (Chen et al., 2025b; Walder & Karkhanis, 2025; Tang et al., 2025).

- **Diverse contextual conditioning**, where the model is provided with various versions of additional context attached to each problem as a way to induce diversity and exploration (e.g. (Chen et al., 2025a) in the context of Lean).

- **Alternative inference schemes** such as agentic interactions with the environment, hierarchical decomposition and multi-agent inference or even tree search in the context of interactive theorem proving (Dong et al., 2024; Gehring et al., 2024; Lample et al., 2022; Gloeckle et al., 2024).

In Section 5, we investigate the effect of these different interventions. For theorem proving with language models returning a response in Lean, a training recipe has evolved that involves separate stages: pretraining; Lean-specific finetuning or distillation; expert iteration, and lastly online reinforcement learning (Lin et al., 2025; Ren et al., 2025). In this work, we focus on mitigating saturation in the last stage: online reinforcement learning. One can look at (Wu et al., 2024a; Ying et al., 2024; Lin et al., 2025; Ren et al., 2025) for more detail on the expert iteration stage in the context of theorem proving in Lean.

## 3 METHOD

### 3.1 HIERARCHICAL LEMMA-BASED PROOF GENERATION

Hierarchical proof generation means decomposing the problem into subproblems which can be solved and evaluated independently. Concretely, we adopt a two-step design where first, a *decomposer* generates a *sketch* which provides a proof provided that the contained lemmas can be proved. The lemmas are then extracted and fed separately to a *prover* with multiple attempts per lemma. This design could be extended into a full recursive strategy where lemmas, too, can be broken down into subgoals further. But we do not employ such a scheme to minimize design choices for the reinforcement learning setup.

In Lean, lemma-based proving hardly comes with a distribution shift compared to whole-proof generation, as logical cuts called `have` statements, and proof omission using the `sorry` tactic can be used to mark such a statement for lemma extraction. Lemma extraction itself can be performed in Lean using the

`extract_goal` tactic: we use the variant that extracts all preceding lemmas as hypotheses to the new lemma, trading expressivity for readability.

Intuitively, lemma-based proving reaps the advantages that encapsulation provides. If parts can be verified independently and sampled more than once, then each part can be optimized separately instead of optimizing the sum of the parts: $\sum_j \max_i x_{ij} \geq \max_i \sum_j x_{ij}$. If there is only one attempt per lemma, the two methods become equivalent.

For reinforcement learning, hierarchical generation comes with more choices than standard whole-proof generation. One could generate $k = 1$ proof attempts per lemma to form a trajectory, assign the overall proof success as reward and apply standard reinforcement learning algorithms, but as explained above that would not deviate from whole-proof generation except for the language model context resets for the individual parts. A natural option, instead, would be to conduct $k > 1$ proof attempts per lemma. In a sketch with $n$ lemmas, this would result in $k^n$ recombined trajectories, which, however, would no longer be independent and could break optimization due to their close correlation. Instead, we propose to **apply a (baselined) policy gradient algorithm at each node in the resulting tree** (concretely, GRPO without $\sigma$-normalization). This means that for the $i$-th proof attempt of lemma $j$, the reward is its proof success $r_i^j$ and its advantage is

$$A_i^j = r_i^j - \frac{1}{k} \sum_{m=1}^{k} r_m^j.$$

For sketches, there are various options:

- Reward as **overall proof success**: $R_o = \min_j \max_i r_i^j$, which depends on $k$.
- Reward as **average proof success**: $R_a = \prod_j \left( \frac{1}{k} \sum_i r_i^j \right)$, which is an unbiased estimator of $\mathbb{E}_{r_1^j \sim d}[\prod_j r_1^j]$, an average independent of $k$.
- Reward as a soft interpolation of overall and average proof success.

We opt for $R = R_o$ exclusively and use $k = 4$ proof attempts per lemma.

Moreover, we apply a **mean baseline to $R$ over $N$ independent sketch attempts**, i.e. given $R_1, \ldots, R_N$, we set the advantage $A_i = R_i - \frac{1}{N} \sum_{m-1}^{N} R_m$ for $i = 1, \ldots, N$.

Note that a single hierarchical proof generation attempt requires $nk + 1$ language model calls, making pass@$(nk + 1)$ the compute-matched comparison.

## 4 EXPERIMENTAL SETUP

### 4.1 TOY MODEL FOR SATURATION

Our experimental setup is designed to *investigate saturation in reinforcement learning* (Section 2). To enable a controlled study, we make two key decisions: **(1)** We opt for distilling two leading models within the 7B parameter class, deferring considerations of expert iteration. **(2)** We limit our reinforcement learning dataset to 6,000 samples – approximately 1,500 times smaller than the expert iteration dataset used by DeepSeek-Prover-V1.5 (Xin et al., 2024). This is both to model the data-constrained regime that naturally appears in several area such as specific area of research-level mathematics and to facilitate rapid experimental cycles.

The following subsections detail our supervised finetuning setup, the model checkpoints used, and the specifics of our reinforcement learning data.

## 4.2 SUPERVISED FINETUNING

As a starting point for reinforcement learning, we perform sequence distillation from two state-of-the-art 7B parameter models: STP (Dong & Ma, 2025), derived from DeepSeek-Prover-V1.5-SFT 7B (Xin et al., 2024), and GoedelProver (Lin et al., 2025), derived from DeepSeek-Prover-V1.5-RL 7B. We intentionally exclude distillation from larger models such as DeepSeek-Prover-V2 671B (Ren et al., 2025), Goedel-Prover-V2 32B (Lin et al., 2025), and their distilled variants, to maintain a controlled experimental environment.

Our finetuning data consists of: proofs from the above mentioned models, curated Lean 4 code in file-format and traced into state-tactic sequences, alongside natural language mathematical problem solving and statement autoformalization data (see Appendix B). Additionally, we include a small portion of proof sketching data obtained from rejection sampling to warm-start the new lemma-based proof style described above.

## 4.3 DATA

We focus on the task of *proof autoformalization*, i.e. formal theorem proving conditioned on a proof given in natural language due to its high relevance for the community and its convenience for sketching. This setup allows us to focus on the Lean proficiency of the model without separately optimizing natural language mathematical reasoning skills simultaneously.

Concretely, we pair up natural language solutions from NuminaMath 1.5 (LI et al., 2024) to the problems in Goedel-Pset-v1, and use the resulting proof autoformalization dataset as our reinforcement learning problem set. We evaluate on ProofNet (Azerbayev et al., 2023) and miniF2F (Zheng et al., 2021), for which we likewise generated natural language solutions.

## 5 RESULTS

Our experiments establish that our initial SFT checkpoint provides a performant starting point for analyzing reinforcement learning with Lean models (Section 5.1), that GRPO indeed suffers from saturation that cannot be remedied with standard techniques (Section 5.2), that sketch-based proving shows appealing initial results in terms of exploration and policy diversity (Section 5.3) and results in competitive benchmark scores (Section 5.4).

## 5.1 PERFORMANT INITIAL CHECKPOINT

Evaluating the initial supervised finetuning checkpoint described in Section 4.2, we obtain scores of 18.7% on ProofNet (valid and test) pass@32 with standard inference, of 16.8% pass@32 with sketching; of 63.5% pass@32 on miniF2F-valid with standard inference and of 50.8% pass@32 with sketching. In all of these evaluations, we couple the neural model with the Lean automation tactic described in Section 3.

These numbers come close to state-of-the-art numbers for 7B parameter models, supporting our claim that the SFT model can be used as a performant initial checkpoint to study saturation in the subsequent reinforcement learning stage.

## 5.2 SATURATION WITH GRPO VARIANTS

Standard reinforcement learning with GRPO plateaus early in our RL setup: as can be seen in Figure 1, the GRPO baseline quickly reaches a ceiling after which few additional problems from the training dataset are solved, while entropy decays. Evaluation scores peak early in training and then subsequently decline. This cannot be remedied with the standard mitigations described in Section 2.3: increasing the dataset size ten-fold and switching to pass@k training both show the same behavior on evaluations and similar saturation slopes

in cumulative pass rate. Pass@k training maintains a higher entropy, but this does not translate to higher cumulative solve rates. (Note that cumulative solve rates for the larger dataset are not comparable with the ones on the default dataset, but the saturation behavior is the same.)

### 5.3 SKETCH-BASED PROVING

For sketch-based proving, we obtain promising for its utility in theorem proving in Lean: According to Figure 1, sketch-based proving reaches **higher cumulative solve rates** than any of the baseline methods, attesting continued exploration and suggesting usefulness in large-scale reinforcment learning setups beyond the scale tested here. **Entropy is maintained higher** than for both of the GRPO runs, coming close to the entropy of pass@k training. In terms of evaluation results, performance starts lower since the initial model was hardly trained on this inference scheme, but mostly catches up with the baselines over the course of training.

Analyzing the scaling behavior at inference time across training steps in Figure 2, we find that standard inference is hardly optimized further over the course of training while sketch-based proving continues to improve from its low starting point.

### 5.4 FINAL BENCHMARK RESULTS

We evaluate the final sketch-based reinforcement learning checkpoint (step 10k) on proof **autoformalization** on miniF2F-test and ProofNet-test. Specifically, we generate 5 proofs per problem with Gemini 2.5 Pro and GPT-5 nano, and add a human-written ground truth proof from the dataset. For each attempt, we select one out of these 11 proofs per problem as input for the sketch-based autoformalizer, i.e. pass@k is computed over the mixed dataset of natural language proofs.

According to the results in Table 1, our proof autoformalizer performs on par with state-of-the-art theorem proving models despite using a short reinforcement learning over 10,000 steps based on only 6,000 training samples. We are not aware of other proof autoformalization models that produce a proof conditioned on a user-provided natural language proof which we could use as baselines.

Moreover, only GoedelProver-SFT and STP are genuine 7B parameter models, while DeepSeek-Prover-V2 7B is distilled from a 671B parameter mixture-of-experts model.

## 6 RELATED WORK

**DeepSeek-Prover-V2** (Xin et al., 2024) uses lemma-based proving for generating a warm-start dataset, but does not use lemma-based proving during reinforcement learning.

**SeedProver** (Chen et al., 2025a) alleges using reinforcement learning with both iterative and lemma-based inference schemes but does not disclose further details.

**Draft-Sketch-Prove** (Jiang et al., 2023) is foundational for proof autoformalization and uses sketches, but relies on classical automation for lemma proofs due to the unavailability of adequate proof generation models at the time.

Dong et al. (2024) apply reinforcement learning to lemma-based proving in Isabelle, arriving at marginal improvements over the baseline which may be linked to the intricacy of optimizing the proposed proof tree generation using a learned value function.

Zhao et al. (2023) rewrite Isabelle proofs in a lemma-based way using LLMs to generate a finetuning dataset.

Wang et al. (2024) uses a recursive lemma decomposition method, trained by supervised finetuning.

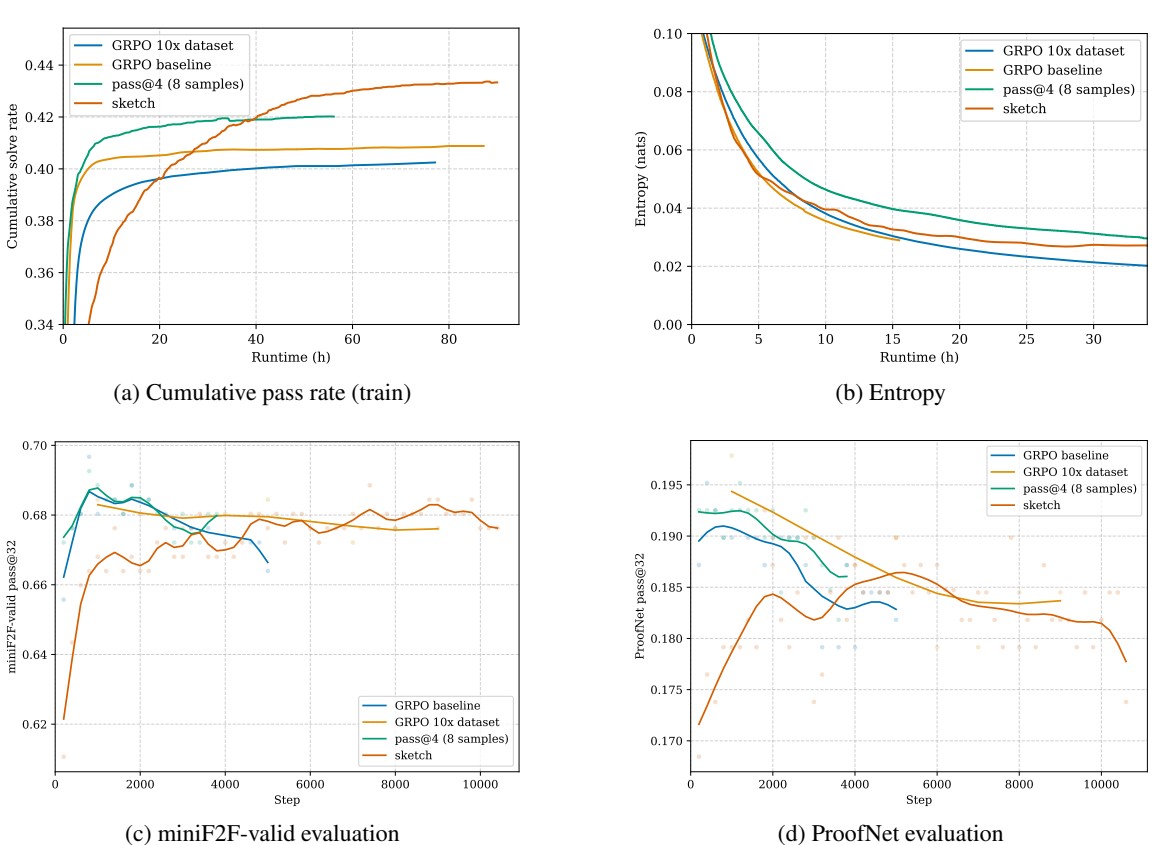

Figure 1: **Reinforcement learning with hierarchical inference shifts plateaus.** **(a)** The fraction of training problems cumulatively solved over the course of training flattens off early with GRPO variants, but continues increasing with sketch-based proving, **(b)** which maintains a higher entropy similar to pass@k training. **(c, d)** Evaluation scores on miniF2F and ProofNet peak early in training for GRPO variants. With sketch-based proving, they start significantly lower but partially catch up later in training.

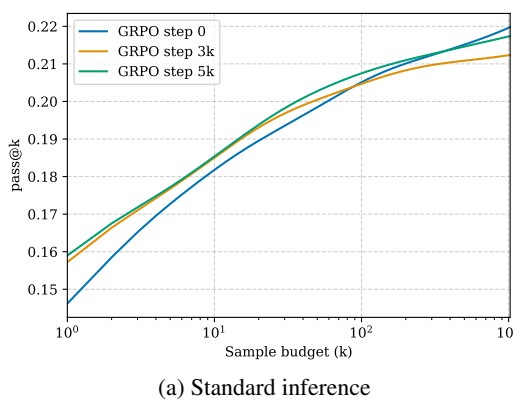 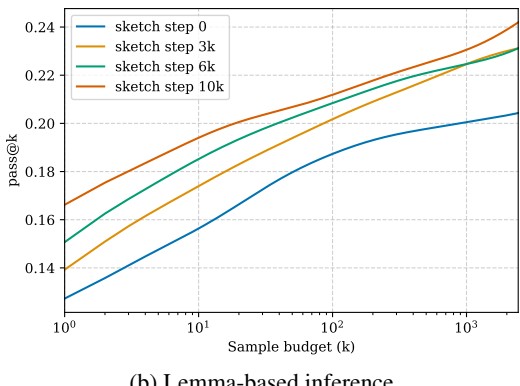

(a) Standard inference                    (b) Lemma-based inference

Figure 2: **Inference scaling on ProofNet-test with standard and lemma-based inference.** GRPO training is unable to improve the model upon pass@1024 evaluations. Sketch-based RL starts at lower numbers and improves to up to a similar performance as standard training. Note that a sketch-based attempts consists of several language model calls, i.e. sample budgets are not comparable. Lemma-based inference uses generated natural language proofs. A similar scaling graph for miniF2F-test with sketch-based proving can be found in Figure 3.

| Prover | Sample budget (#Attempts) | Sample budget (#LLM calls) | ProofNet-test | MiniF2F-test |
|---|---|---|---|---|
| Goedel-Prover-SFT 7B | 32 | – | 15.6% | 57.6% |
| (Lin et al., 2025) | 3200 | | – | 62.7% |
| STP 7B (Dong & Ma, 2025) | 128 | – | 18.0% | 57.2% |
| *without ProofNet-valid, miniF2F-valid* | 3200 | – | 23.1% | 61.1% |
| DeepSeek-Prover-V2 7B | 32 | – | 23.0% | 75.6% |
| (CoT, Xin et al. (2024)) | 128 | – | 25.4% | – |
| | 1024 | – | 29.6% | 79.9% |
| | 8 | 59 / 69 | 19.2% | 55.7% |
| | 32 | 237 / 278 | 20.4% | 58.2% |
| Ours 7B (sketch autoformalization) | 128 | 947 / 1111 | 21.4% | 59.9% |
| | 512 | 3787 / – | 22.5% | – |
| | 2048 | 15146 / – | 23.9% | – |

Table 1: **Results on miniF2F and ProofNet autoformalization, compared to various theorem probing methods.** Note that unlike other methods, our model has access to ground-truth and model-generated natural language solutions to evaluate proof autoformalization performance. Removing access to ground truth and only using model-generated natural language solutions does not change significantly the numbers. One sketch attempt corresponds to several language model calls depending on the number of lemmas in each sketch: we report the equivalent sample budget as $nk + 1$ for ProofNet and miniF2F, respectively, in the central column.

## 7 CONCLUSION AND OUTLOOK

For scaling formal theorem proving to problems beyond mathematical competitions, test-time inference needs to be scaled by large amounts. While vanilla whole-proof generation suffers from exponential decay of proof

rates with increasing proof complexity, lemma-based and iterative proof generation provide alternatives with better scaling behaviors.

In this work, we explored in a well-controlled setting how online reinforcement learning combined with hierarchical decomposition can improve performances and tackle the limitations of whole-proof generation methods. While not yet definitive, the evidence presented in this article suggests the strong potential of these hierarchical approaches and sketch-based proofs in formal mathematics.

## REPRODUCIBILITY STATEMENT

Our approach, and in particular the adaptations to previous reinforcement learning approaches are detailed in Section 3. The data used to obtain the initial SFT checkpoint are detailed in Appendix B, while the data later used for training are specified in Section 4.3. The evaluation datasets can be found at yangky11 & contributors (2025) for MiniF2F, with additionnal natural language solutions coming from (Research & contributors, 2022) and Azerbayev & contributors (2023) for ProofNet.

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

## A LSP ENVIRONMENT

We use a custom Lean environment that interacts with the Lean language server. This allows for fast interaction with Lean due to caching of imports. Lemma extraction is implemented using extract_goal.

## B  DETAILS ON THE FINETUNING DATASET

Following Gu et al. (2025), we collect a finetuning dataset consisting of roughly 1B Lean tokens. The training data spans a diverse set of math and Lean-related tasks and includes the following:

- **Theorem proving**: conjectures and proofs from STP (Dong & Ma, 2025), and solution autoformalization data from the Goedel-Pset-v1-Solved dataset obtained by mapping Lean solutions with natural language solutions.

- **Natural language problem solving**: problems and solutions in natural language from the NuminaMath-1.5 dataset (LI et al., 2024).

- **Lean code**: Lean 4 code curated from GitHub, filtering with GPT-4o and manual heuristics to ensure only Lean 4 examples are included.

- **Auto-formalization**: mathematical statements in natural language paired up with their Lean counterparts, from the following high-quality datasets: CombiBench (Liu et al., 2025), Compfiles, FormalMATH (Yu et al., 2025), Goedel-Pset (Lin et al., 2025), Lean Workbook (Ying et al., 2024), miniF2F (Zheng et al., 2021), ProofNet (Azerbayev et al., 2023), and PutnamBench (Tsoukalas et al., 2024).

- **Tactic prediction**: Lean repositories traced at the tactic level into (state, tactic, next state) triples. We use data from LeanUniverse (Aram H. Markosyan, 2024) and extract additional data using the Pantograph (Aniva et al., 2025) tool from the STP dataset.

## C  AUTOFORMALIZATION INFERENCE SCALING ON MINIF2F-TEST

We provide the analobon to Figure 2 for miniFF-test in Figure 3.

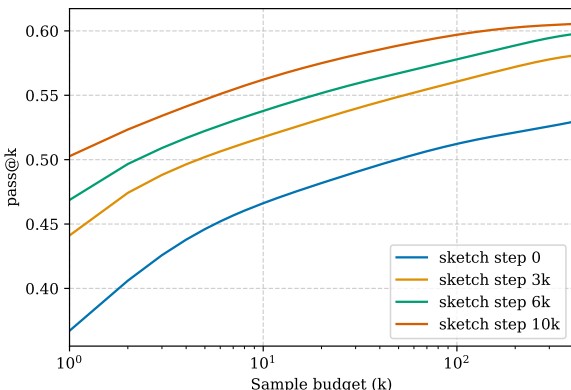

Figure 3: **Autoformalization inference scaling on miniF2F-test with sketch-based inference.** We use ground truth proofs and generated proofs, and pass@k is evaluated over $k$ attempts that may stem from different proof sources.

