# OpenReview forum: "Reinforcement learning for hierarchical proof generation in Lean 4"
_ICLR.cc/2026/Conference — Submitted to ICLR 2026_

### Official Review · Reviewer_X1h4 · 2025-10-28

**Soundness:** 1
**Presentation:** 2
**Contribution:** 2
**Rating:** 2
**Confidence:** 4

**Summary:**

This paper investigates the combination of online reinforcement learning (RL) with hierarchical, lemma-based proof generation for automated theorem proving in Lean 4. The authors identify two primary challenges with current methods:
- Standard "whole-proof" generation suffers from exponentially decreasing success rates as proof length increases.
- RL-based training in data-scarce domains like formal mathematics quickly "saturates," leading to performance plateaus.

The authors design a controlled experiment using a small 7B parameter model and a deliberately data-constrained setting (only 6,000 RL training samples) to induce saturation. They compare their hierarchical "sketch-based" RL approach against standard whole-proof generation baselines (GRPO). The results show that while the GRPO baselines quickly plateau in both cumulative training solves and evaluation performance, the hierarchical method continues to improve, avoids saturation, and maintains higher entropy.

**Strengths:**

- The paper clearly articulates two significant and well-known problems: the scalability of long-form proof generation and the saturation of RL in data-constrained environments.

**Weaknesses:**

- Clarity and Experimental Setting: The paper's narrative and experimental design are unclear. The method is not clearly explained; despite reading the methodology section repeatedly, I was unable to fully grasp the approach. Furthermore, the use of limited training data to mimic data scarcity (as in Lean/other environments) seems questionable or an odd choice for demonstrating the method's effectiveness.

- Doubtful Main Results: The main experimental results presented in Table 1 are questionable. All baseline methods are designed for the theorem-proving task. This design inherently gives the sketch methods the advantage of having a natural language proof in advance. The authors claim the performance of the sketch methods without natual language proof is close to their proposed method but never show these results in the paper.

- Misleading Figures: The figures in the paper are somewhat misleading. Figures 1(c) and 1(d) use a smoothed line to suggest the superiority of the sketch method. However, the best result actually comes from the GRPO Baseline in the early steps. The sketch method never clearly surpasses the baseline in either the miniF2F or ProofNet environments across the entire process shown.

Overall, these are just a few of the listed weaknesses; the paper's results are highly doubtful, and I do not recommend this paper for acceptance.

**Questions:**

- In Figures 1(a) and (b), it's recommended to use RL iterations instead of hours to show the progress of the training. And why in figure (b) the time is less compared to the time in (a)?

- It would be better to have a figure illustrating your methodology or a pseudo-code. To clearly illustrate the pipeline of your method. And the overall writing and storytelling should be improved significantly

- Personally speaking, the point of the sketch method can prevent saturation in RL is really weird, since the sketch method has a much lower starting point compared to the GRPO baseline. And normally, when tuned properly, it's not that easy to fail for GRPO in formal tasks. (As we can see in Godel Prover-v2, Kimina-Prover, and Seed Prover)

---

### Official Review · Reviewer_L8s3 · 2025-10-31

**Soundness:** 2
**Presentation:** 1
**Contribution:** 1
**Rating:** 0
**Confidence:** 4

**Summary:**

The paper studies how online reinforcement learning can be coupled with a hierarchical (lemma-based) style of proof generation. The authors report that hierarchical proof generation performs favorably compared to standard whole-proof generation, showing higher sample efficiency and plateauing later in reinforcement learning runs.

**Strengths:**

- The hierarchical design of rewards makes sense in the context of interactive theorem proving, especially when one adopts a sketch-based proof generation strategy.

**Weaknesses:**

- The presentation is so confusing that it’s difficult to understand how exactly the authors implemented their framework and carried out the experiments. For example, in the sketch-based experiment, no detail of the model or settings were given. How are drafts or proof skeletons generated? By what model, trained or fine-tuned on what dataset?
- In general the paper also lacks proper mathematical formalization of the approaches. How exactly are the hierarchical rewards integrated into GRPO? In such a setting, what are the actions and states in terms of an MDP?

Minor: various places where the English is incomprehensible. For example, on line 241: “ .. we obtain promising for its utility in theorem proving in Lean”

**Questions:**

What is the exact setting of the “sketch-based” proving / experiments? What is the decomposer model and what is the prover model? How are they related to the models in section 4.2?

---

### Official Review · Reviewer_byhV · 2025-11-01

**Soundness:** 3
**Presentation:** 3
**Contribution:** 2
**Rating:** 4
**Confidence:** 4

**Summary:**

This paper investigates combining online reinforcement learning with hierarchical (lemma-based) proof generation in Lean 4. The authors propose a two-step approach where a decomposer first generates a proof sketch containing lemmas, which are then individually proved by a prover model. They train this hierarchical system using GRPO-based reinforcement learning applied at each node in the proof tree. The key findings are: (1) hierarchical proof generation can be trained with RL using minimal supervised data, (2) it shows higher sample efficiency and delays plateaus compared to standard whole-proof GRPO training, and (3) it achieves competitive benchmark results despite using only 6,000 training samples and 10,000 RL steps. The approach is evaluated on proof autoformalization tasks using miniF2F and ProofNet benchmarks.

**Strengths:**

1. Originality: This work is the first systematic study combining RL with hierarchical proving in Lean 4, filling a gap between prior inference-only hierarchical methods and RL-only whole-proof approaches. The application of policy gradients at each node in the proof tree rather than treating the entire generation as a single trajectory is creative and the experimental design focusing on the data-constrained regime is thoughtful.

2. Quality: Solid experimental methodology with appropriate baselines (GRPO variants, pass@k training) and proper use of established benchmarks. Clear identification of saturation criteria and systematic measurement and honest acknowledgment of limitations (small scale, "toy model").

3. Clarity: Well-motivated problem statement with clear exposition of the scaling challenge (p^n success rate). Good visual presentation (Figure 1 effectively shows the saturation phenomenon). Appropriate background section establishing key concepts and clear description of the two-step hierarchical approach.

4. Significance: This work addresses practically important problem of scaling to complex proofs, and demonstrates feasibility of training hierarchical decomposition policies with RL.

**Weaknesses:**

Major:

1. Unfair computational comparison

Hierarchical inference requires nk+1 LLM calls per "attempt" while standard generation requires 1. This makes all comparisons fundamentally unfair: the hierarchical method simply uses more compute. The paper compares "pass@32" for standard vs. "pass@8" for hierarchical, but the hierarchical version uses 59-69 LLM calls (Table 1), making it actually pass@(59-69). A fair comparison would be: standard pass@60 vs. hierarchical pass@8, which is not shown. This undermines the main claim of "higher sample efficiency"

2. Limited experimental scale

Only 6,000 RL training samples (1,500x smaller than DeepSeek-Prover-V1.5), 10,000 RL steps, and 7B parameter models tested. These choices may artificially favor hierarchical methods, perhaps standard GRPO plateaus simply because the dataset is too small. The authors acknowledge this as a "toy model" but then draw general conclusions.

3. Missing ablations and baselines

3.1 No ablation on k (number of attempts per lemma), what if k=1 or k=8?

3.2 No comparison to the simplest baseline: just use more samples with standard generation.

3.3 No ablation on reward choice ($R=R_o$ vs. $R=R_a$), why this choice?

3.4 No experiments with larger datasets to see if standard GRPO would also avoid plateaus.

3.5 No testing at larger model scales (13B, 32B).

4. Unclear generalization

Results are on proof autoformalization (conditioned on NL proofs), which is different from de novo theorem proving. The small-scale "toy model" may not represent behavior at realistic scales. The saturation phenomenon observed may be specific to this small dataset size.

Minor:

5. Incomplete technical details:

Learning rates not specified (important for RL); batch sizes not given; number of gradient update steps unclear. Wall-clock time comparisons missing (how much slower is hierarchical inference?). No error bars or confidence intervals despite stochastic training.

6. Presentation:

The advantage formula in Section 3.1 could be clearer. Missing details on how lemmas are extracted programmatically.
No examples of actual generated proof sketches. The term "saturation" is used loosely before being defined.

7. Limited analysis:

There is no analysis of what makes a "good" decomposition. No investigation of failure modes, or analysis of lemma complexity distribution.
Limited discussion of when hierarchical methods help vs. hurt. No comparison of proof lengths between methods.

8. Claims:

"Strong numbers on evaluations" (abstract), the results are competitive but not state-of-the-art.
"Outperforms standard whole-proof generation". this seems to be only true with the unfair computational budget comparison.
Claims about "sample efficiency" are misleading given the LLM call discrepancy.

**Questions:**

Critical:

1. Fair computational comparison: Could you please provide results for standard GRPO with pass@60 (or the number of LLM calls that matches your hierarchical pass@8)?

2. Scaling behavior: Have you tested whether standard GRPO also avoids plateaus with larger datasets (e.g., 60,000 samples instead of 6,000)? Perhaps the issue is dataset size, not the inference method?

3. Ablation on k: What happens with k=1 (hierarchical structure but no extra samples per lemma) vs. k=8? This would isolate the benefit of decomposition from simply using more samples.

4. Baseline comparison: What if you just do standard GRPO but with 5x more samples per problem (to match the ~5x LLM calls of hierarchical)? Does it also avoid plateaus?

Important:

5. Reward choice: Why did you choose $R=R_o$ (overall proof success) instead of $R=R_a$ (average proof success)? Did you try both?

6. Larger scales: Have you tested this at 13B or 32B parameter scales? Do the benefits still hold?

7. Have you investigated if this approach work for theorem proving without natural language proof hints?

8. How do you ensure the generated decompositions are "good"? Can you show examples of learned decomposition strategies?

9. Failure analysis: What fraction of sketches contain 0 lemmas? What is the distribution of lemma counts? When does hierarchical generation hurt?

10. Can you show training curves (loss, KL divergence, etc.) not just evaluation metrics?

Minor Questions:

11. Implementation: What are the exact learning rates, batch sizes, and number of gradient steps used? How much slower is hierarchical inference in practice (wall-clock time)?

12. Lemma extraction: Can you provide more details on the extract_goal tactic and show examples?

---

### Meta-Review · Area_Chair_qYVF · 2026-01-04

**Summary:**

The reviewers unanimously and strongly recommend rejection (average score 2), mainly citing:
- poor presentation, in both reviews mathematical and narrative style (all reviewers);
- missing details (all reviewers) and analyses (byhV and X1h4);
- unfair experimental settings or improper comparisons to prior work (byhV and X1h4).

I agree with the reviewers' recommendation.

*Content aside, the paper violates the formatting guidelines (margins), which should alone be grounds for rejection.*

**Reviewer Concerns:**

N/A (no response from authors)

**Reviewer Scores:**

N/A (no response from authors)

---

### Decision · Program_Chairs · 2026-01-26

Reject